# HPMCP-Coated Microcapsules Containing the Ctx(Ile^21^)-Ha Antimicrobial Peptide Reduce the Mortality Rate Caused by Resistant *Salmonella* Enteritidis in Laying Hens

**DOI:** 10.3390/antibiotics10060616

**Published:** 2021-05-21

**Authors:** Cesar Augusto Roque-Borda, Larissa Pires Pereira, Elisabete Aparecida Lopes Guastalli, Nilce Maria Soares, Priscilla Ayleen Bustos Mac-Lean, Douglas D’Alessandro Salgado, Andréia Bagliotti Meneguin, Marlus Chorilli, Eduardo Festozo Vicente

**Affiliations:** 1School of Agricultural and Veterinarian Sciences, São Paulo State University (Unesp), Jaboticabal, São Paulo 14884-900, Brazil; cesar.roque@unesp.br; 2School of Sciences and Engineering, São Paulo State University (Unesp), Tupã, São Paulo 17602-496, Brazil; larissa.pereira@unipac.com.br (L.P.P.); priscilla.mac-lean@unesp.br (P.A.B.M.-L.); douglas.salgado@unesp.br (D.D.S.); 3Poultry Health Specialized Laboratory, Biological Institute, Bastos, São Paulo 17690-000, Brazil; elisabete.guastalli@sp.gov.br (E.A.L.G.); updbastos@biologico.sp.gov.br (N.M.S.); 4School of Pharmaceutical Sciences, São Paulo State University (Unesp), Araraquara, São Paulo 14801-902, Brazil; andreia.meneguin@unesp.br (A.B.M.); marlus.chorilli@unesp.br (M.C.)

**Keywords:** AMP, HPMCP, chicks, microencapsulation, mortality rate

## Abstract

The constant use of synthetic antibiotics as growth promoters can cause bacterial resistance in chicks. Consequently, the use of these drugs has been restricted in different countries. In recent years, antimicrobial peptides have gained relevance due to their minimal capacity for bacterial resistance and does not generate toxic residues that harm the environment and human health. In this study, a Ctx(Ile^21^)-Ha antimicrobial peptide was employed, due to its previously reported great antimicrobial potential, to evaluate its application effects in laying chicks challenged with *Salmonella* Enteritidis, resistant to nalidixic acid and spectinomycin. For this, Ctx(Ile^21^)-Ha was synthesized, microencapsulated and coated with hypromellose phthalate (HPMCP) to be released in the intestine. Two different doses (20 and 40 mg of Ctx(Ile^21^)-Ha per kg of isoproteic and isoenergetic poultry feed) were included in the chick’s food and administered for 28 days. Antimicrobial activity, effect and response as treatment were evaluated. Statistical results were analyzed in detail and indicate that the formulated Ctx(Ile^21^)-Ha peptide had a positive and significant effect in relation to the reduction of chick mortality in the first days of life. However, there was moderate evidence (*p* = 0.07), not considered statistically significant, in the differences in laying chick weight between the control and microencapsulation treatment groups as a function of time. Therefore, the microencapsulated Ctx(Ile^21^)-Ha antimicrobial peptide can be an interesting and promising option in the substitution of conventional antibiotics.

## 1. Introduction

*Salmonella* is a bacterium of public health importance and can contaminate food or spaces due to its high risk of transmission, mainly by its common host, the poultry [1]. The frequent transmission, as well as in attempts to control *Salmonella*, allowed this microorganism to generate or acquire resistance to several commercial drugs. *Salmonella-*accelerated proliferation would be related to the immunity alteration by stress factors, products of excessive manipulation or environmental conditions [2]. In 2017, in the last update of the WHO global warning about bacterial resistance, a list of global priorities of resistant bacteria to antibiotics was declared and published, in which *Salmonella* sp. fluoroquinolone-resistant was classified in the high priority group (number 2). Therefore, based on public health policies, there is a very high degree of concern about these aggressive pathogenic bacteria species. In this scenario, antibiotics used in the poultry industry are increasingly restricted and discovery of new drugs is becoming essential and in urgent demand [3,4].

In recent years, antimicrobial peptides (AMPs) have been a central object of study, attributable to their great capacity to control bacterial pathogens, including viruses and fungi [5,6]. Specifically, the Ctx(Ile^21^)-Ha antimicrobial peptide is an amphipathic and cationic peptide, isolated from an Brazilian amphibian skin (*Hypsiboas albopunctatus*) [7], which has a high antimicrobial capacity, demonstrated in pathogens of public health interest [8]. Thus, Ctx(Ile^21^)-Ha and others AMPs are considered natural antibiotics, as they are part of a biological defense innate immune system. In addition, they are biocompatible, can modulate immune systems and have high biological activities with minimal concentrations [9]. An interesting feature of AMPs is that they can generate a minimal level of bacterial resistance [10]. As a result of these attractive characteristics, AMP application in poultry as a feed additive is promising, but challenging [11,12,13,14].

Although there is optimistic application, the use of these molecules is limited due to instability factors, such as denaturation or acid hydrolysis degradation, produced by gastric acids in the stomach of monogastric animals. To overcome these issues, coated bioformulations to protect bioactive molecules are demanded. Microencapsulation, a standard pharmacotechnical methodology and a very well-established technique in the literature, is used to control, protect and maintain compounds’ biological activities [15].

Some types of encapsulations were developed to improve poultry production. For example, spray drying is employed to microencapsulated probiotics and can maintain 90% of stability, allowing them to be installed in the chicken intestine [16]. Enteric coating is a protection method widely used in pharmaceuticals, which permits the targeted transport of biomolecules or drugs to be released at a specific site, depending on the conditions of the polymer used, such as hydroxypropyl methylcellulose phthalate (HPMCP) [17]. This is a modified polymer, derived from cellulose and is pH dependent, which it tends to dissolve in liquid solutions at pH > 6.5, playing an excellent role as a drug carrier against intestinal pathogens [18].

These parameters allowed us to design an innovative product based on microencapsulates and enteric coating of a biocompatible molecule with potential antimicrobial activity. The Ctx(Ile^21^)-Ha AMP was chosen due to its properties, previously reported by our research group [8]. In this way, the objective of this study was to evaluate the in vivo effect of HPMCP-coated microcapsules containing the Ctx(Ile^21^)-Ha antimicrobial peptide application against *Salmonella* Enteritidis in chickens, to demonstrate its great potential as an innovative natural feed additive in poultry production.

## 2. Material and Methods

### 2.1. Chemical Reagents

HPMCP (Grade HP-55, Nominal Phthalyl Content 31%) was kindly donated by Shin-Etsu Chemical (Tokyo, Japan), and the other chemical reagents were obtained in HPLC grade (Sigma-Aldrich Co., Missouri, USA). *N,N*-dimethylformamide (DMF) was purchased from Neon Comercial (São Paulo, Brazil), dichloromethane (DCM) was purchased from Anidrol Products Laboratories (São Paulo, Brazil), sodium alginate with low molecular weight (12,000–40,000 g mol^−1^, M/G ratio of 0.8) and aluminum chloride were obtained from Êxodo Científica (São Paulo, Brazil). Fmoc-amino acids were purchased from AAPPTEC (Kentucky, USA). Brain Heart Infusion (BHI) broth, Mueller Hinton (MH) agar, Bright Green Agar (BG), selenite broth (SB), nutrient broth (NB), and other microbiological reagents were purchased from SPLABOR (São Paulo, Brazil).

### 2.2. Ctx(Ile^21^)-Ha Antimicrobial Peptide Synthesis

The antimicrobial peptide Ctx(Ile^21^)-Ha was synthesized manually using solid phase peptide synthesis (SPPS) with a Fmoc strategy protocol. The complete methodology is described according to Roque Borda et al. [19] Briefly, peptide was assembled at a 0.2 mmol scale on a Fmoc-Rink Amide resin of 0.68 mmol g^−1^ substitution, using three-fold excess and preconditioned for 15 min in DMF and DCM as main SPPS solvents. 4-methylpiperidine/DMF (1:4, v/v) was used to remove the Fmoc amino group protectors from amino acids. Having finished the entire peptide primary sequence, Ctx(Ile^21^)-Ha peptide was separated from the resin using a solution containing trifluoroacetic acid/ultrapure water/triisopropylsilane (95:2.5:2.5, *v*/*v*/*v*), at 160 rpm for 2 h at room temperature. Next, samples were freeze-dried (Liotop model K108, Sao Paulo, Brazil) to obtain the peptide in a white and flocculent powder material.

The peptide purity degree was determined by analytical HPLC (Shimadzu, model Prominence with membrane degasser DGU-20A5R, UV detector SPD-20A, column oven CTO-20A, automatic sampler SIL-10AF, fraction collector FRC-10A and LC-20AT dual-pump, C18 column) at a flow rate of 1 mL min^−1^ and a detection at wavelength of 220 nm, using as mobile phases 0.045% aqueous TFA (eluent A) and 0.036% TFA in acetonitrile (eluent B) for 30 min. Subsequently, samples were lyophilized and stored until use. The Ctx(Ile^21^)-Ha peptide was employed only if the purity degree was higher than 95%. After that, the peptide was confirmed and characterized by ESI-MS (Electron Spray Injection Mass Spectrometry), employing a mass spectrometer (Bruker, CA, USA). Pure Ctx(Ile^21^)-Ha peptide concentrations were determined by UV spectroscopy, considering tryptophan extinction coefficient of 5600 M^−1^ cm^−1^ at a wavelength of 220 nm.

### 2.3. Development of Ctx(Ile^21^)-Ha Coated Microcapsules (ERCtx)

Ctx(Ile^21^)-Ha was encapsulated by an ionotropic gelation method, following the method described in Roque-Borda et al. [19] Summarily, the peptide-alginate solution was prepared with an initial concentration of 14 (PEP1) and 28 µmol L^−1^ (PEP2) of Ctx(Ile^21^)-Ha peptide in 2% (w/w) sodium alginate, homogenized using an UltraTurrax-T18 (IKA-Labortechnik, Staufen, Germany) at 25,000 rpm min^−1^ and sonicated with an ultrasound probe (Hilscher, Hesse, Germany) for 15 min. Therefore, a crosslinking solution was prepared with 5% aluminum chloride. Capsules were obtained using a syringe pump (NE-1000, New Era Pump System Inc., New York, USA) with a feed flow rate of 1.5 µL h^−1^ at room temperature. After that, they were dried and stored in darkness.

Ctx(Ile^21^)-Ha microcapsules were coated by the fluidized-bed method, preparing a coating solution with 10% w/w HPMCP, 25% w/w ammonium hydroxide, 2.5% w/v triethylcitrate and 62.5% of water. The microcapsules were placed on a fluidized-bed (LabMaq MLF 100, Sao Paulo, Brazil) at 40 °C, 0.25 L min^−1^ blower, 0.4 mL min^−1^ peristaltic pump and 100% vibration as a system condition and yielded a 75% of peptide microencapsulation.

### 2.4. In Vivo Experiment in Chicks

Animal experiments were approved by the local Animal Ethics Committee-School of Sciences and Engineering, UNESP, Tupã, Brazil (Number process. 06/2018-CEUA). The mortality rate of the chicks was the guiding variable for the calculation of the sample size [20].

To perform the in vivo assays, 135 commercials female chicks from Hy-lines Brown, Brazil, were acquired from a commercial hatchery. Chick swabs were taken at random, and a box swab sample, to detect *Salmonella* Enteritidis (*S.* Enteritidis) in newborn chickens and verify that they were free of infection. Thus, confirming that all the chicks used in this experiment were negative for this bacterium, the samples were cultured in SB 2X for 24 h at 37 °C. For the inoculum, *Salmonella* Enteritidis resistant to nalidixic acid and spectinomycin (SE Nal^R^Spc^R^, code P125109-bacterial strain from donated by the Laboratory of Ornithopathology FCAV/UNESP), was grown in a nutrient broth (NB) for 24 h at 37 °C. All chicks challenge was carried out with using 0.2 mL of 10^9^ CFU mL^−1^ of *S.* Enteritidis.

Chicks were randomly distributed into three groups, separated into 45 chicks for each treatment. They were identified with enumerated tape around the right leg. From the first day of the experiment, animals received water and powder feed ad libitum and doses of antimicrobial peptide Ctx(Ile^21^)-Ha microencapsulated were added to the feed and administered to chicks from the first day of life. Control treatment (CTRL) was defined as that which received only the initial commercial feed for chicks without any additives; the PEP1 treatment received the ERCtx with 20 mg of Ctx(Ile^21^)-Ha microencapsulated per kg of poultry feed and the PEP2 treatment received the ERCtx with 40 mg of Ctx(Ile^21^)-Ha microencapsulated per kg of poultry feed. Both microparticles were added to the initial commercial feed of the control treatment (isoproteic and isoenergetic for chicks, in the first 28 days of life, considering a mean of total amount of accumulated poultry feed of 598.5 g consumed, according to the management guide of Hy-line Brown commercial laying hens).

For the chick cloacal swab, 15 chicks were selected for each group. A collection of the fecal excretion was performed two times each week. The collected samples were incubated in 3 mL of SB and Novobiocin (Nov) at 37 °C for 24 h, to later be seeded in BG Nal/Spec and incubated again. This procedure was repeated throughout the experiment. The results were expressed as presence/absence of *S.* Enteritidis, depending on their being positive or negative for the pathogen, respectively [21,22]. In addition, chicks were weighed alive from 12 days of age until the end of the experiment. For the microbiological analyses, five chicks were used for each treatment for the day of the analyses (total of 30 chicks per treatment). Likewise, the chicks were weighed in triplicate for each treatment.

For the evaluation of intestinal infection, five chickens from each group were sacrificed for the count of *S.* Enteritidis in the cecal content, carried out on days 2, 5, 7, 14, 21 and 28 post-infection (dpi). The samples were collected aseptically, with the help of sterilized forceps and individual scissors for each chick. The previously weighed tubes were conditioned in PBS pH 7.4 in the ratio 1:10, *w*/*v*, and were homogenized in vortex. The samples were seeded and cultured on a BG Nal/Spec agar plate at 37 °C for 24 h and counted in colony-forming units (CFU).

### 2.5. Statistical Analysis

To perform the total count of *S.* Enteritidis in CFU/mL, the data were transformed into a Napierian logarithm (Ln) to adapt the model recommended in ANOVA. Mortality was analyzed using the Chi-square test. The results were analyzed by software R package version 3.6.0 (R Foundation for Statistical Computing: Vienna, Austria).

## 3. Results

### 3.1. Peptide Analysis

Ctx(Ile^21^)-Ha AMP was synthesized successfully by SPPS methodology. In the initial analysis, 590 mg of crude mass of AMP was obtained, which was subsequently purified. The purification yield was 20% with a total pure mass of 120 mg. The characterization analysis was carried out by HPLC and Mass Spectrometry, confirmed the obtaining of Ctx(Ile^21^)-Ha AMP (MW = 2289.72 g mol^−1^), shown in Figure 1.

Ctx(Ile^21^)-Ha AMP was microencapsulated with sodium alginate and coated with HPMCP (Figure 2). The final products (ERCtx) used for in vivo evaluation are represented by PEP1 and PEP2. The microencapsulation development and characterization are described according to Roque-Borda et al. [19].

### 3.2. In Vivo Results

The degree of invasiveness present in this study, together with the ethical requirements in the use of animals in experiments, added to the preservation of the quality of handling, led to the use of 45 animals per treatment, making a total of 135 animals. All the chicks were challenged with *Salmonella* Enteritidis from the first day of life, and the PEP treatment groups received a different dose of AMP (Section 2.4). Weighing difference and mortality rate were evaluated using rigorous statistical analysis described in the Materials and Methods section.

#### 3.2.1. Post-Inoculation Treatment Study 

The mortality results showed the significant (α = 0.05) influence of the application of coated-microparticles loaded with Ctx(Ile^21^)-Ha AMP in the treatment on the registered mortality percentages (*p* =0.03), by using Qui-square test, with two degrees of freedom (df = 2). Therefore, mortality percentages differ significantly between treatments. In the control treatment (CTRL), the estimated risk of death for a chicken (R_CTRL_) corresponds to the probability estimation of chick death, given the non-ingestion of the microparticles with antimicrobial peptide Ctx(Ile^21^)-Ha; that is, R_CTRL_ = 13/45 = 28.89%. In parallel, risk of death for a hen treated with PEP1, which is the estimate of the probability of death of the chick given the ingestion of 20 mg of Ctx(Ile^21^)-Ha microencapsulated per kg of poultry feed, is R_PEP1_ = 4/45 = 8.89%. Finally, the risk of death for a hen treated with PEP2, which corresponds to the estimated probability of death of the chick given the ingestion of 40 mg of Ctx(Ile^21^)-Ha microencapsulated per kg of poultry feed; that is, R_PEP2_ = 6/45 = 13.33%.

The mortality results were explored with the percentage distribution conditioned by each treatment, where the proportion of the results was presented as a function of the corresponding treatment to which the chicks were subjected (Figure 3). Due to the statistics illustrated in Figure 3, the risk of death was compared two by two, using estimated Relative Risk (RR) statistic that quantifies the relationship between higher mortality through the relationship between risks [23], the numerator having the highest risk and the denominator, the lowest risk.

In this study, PEP1 and PEP2 treatments have the function of protecting the chicks from the direct and indirect harmful effects of *S.* Enteritidis inoculation. Therefore, the control group corresponds to the exposure group and, consequently, to a higher risk. Thus, three RR estimates could be produced, but only two were of real interest for analysis.

The first was the RR of mortality among the animals in the CTRL and PEP1 treatments (R_CTRL-PEP1_):(1)$${\hat{\mathrm{RR}}}_{\mathrm{CTRL}-\mathrm{PEP}1}=\;\frac{{\hat{R}}_{\mathrm{CTRL}}}{{\hat{R}}_{\mathrm{PEP}1}}$$

According to the results, a value of 3.25 (*p* = 0.01) was obtained. This implies that the risk of death of the hen is 3.25 times higher in the CTRL condition compared to PEP1.

The second, the estimated relative mortality risk between animals in the CTRL and PEP2 treatment (R_CTRL-PEP2_):(2)$${\hat{\mathrm{RR}}}_{\mathrm{CTRL}-\mathrm{PEP}2}=\;\frac{{\hat{R}}_{\mathrm{CTRL}}}{{\hat{R}}_{\mathrm{PEP}2}}$$
and the following results were obtained: a value of 2.17 (*p* = 0.04) is reached, which indicates that the risk of death of the hen is 2.17 times higher in the CTRL condition compared to PEP2.

Importantly, the use of the hypothesis test (H_0_ or H_1_) performed was one-sided since the treatment is unlikely to increase mortality at a 5% significance level (α = 0.05). Thus, the null hypothesis (H_0_: RR = 1) is rejected in both risk tests (*p* = 0.04) in favor of the alternative hypothesis (H_1_: RR > 1), which allows a 95% Confidence Interval (CI) of Relative Risks as a form of interval estimation for the RR population. This is represented by:(3)$$95\%\;{\mathrm{CI}\;\mathrm{to}\;\mathrm{RR}}_{\left(\mathrm{CTRL}-\mathrm{PEP}1\right)}=\left[1.36,\infty \right)$$

That is, with 95% CI, it is possible to affirm that the true RR in question (the population) is at least greater than or equal to 1.36. This implies that the true mortality in a population is at least 36% higher for control group animals compared to the PEP1-treated group:(4)$$95\%\;{\mathrm{CI}\;\mathrm{to}\;\mathrm{RR}}_{\left(\mathrm{CTRL}-\mathrm{PEP}2\right)}=\left[1.04,\infty \right)$$

Therefore, with 95% CI, it is possible to affirm that the true RR in question (the population) is at least 1.04. This implies that the true mortality in a population is at least 4% higher for the chicks in the control group, compared to the PEP2-treated group.

In this experiment there was no reduction in mortality when the peptide concentration was increased. Therefore, it is not necessary to test the statistical difference in mortality risk between these two doses. Moreover, a higher protection (lower mortality) was obtained with fewer resources (peptide mass), which is important in an industrial approach. That is, due to the results, the reduction in mortality does not improve due to the increase in concentration. However, in the best of cases, it remains the same. Furthermore, it is highlighted that there is statistical evidence that PEP1 treatment reduces total mortality, and not only due to *S.* Enteritidis infection. They can be used to establish a metric that quantifies the protection acquired by chicks, because they were also subjected to a treatment with peptides (PEP1 or PEP2). In addition, this can be due to the nature of the action of antimicrobial peptides, which is to protect the chicks against *S.* Enteritidis (Figure 4).

The Protection Factor (PF) is the statistic that quantifies the reduction in mortality risk due to the use of PEP1 or PEP2. In this case, mathematically, the PF is nothing more than the opposite of RR, shown in Equation (5). Therefore, the estimation of the Protection Factor that PEP1 treatment has on mortality, when compared with the CTRL, is given by:(5)$${\hat{\mathrm{PF}}}_{\left(\mathrm{CTRL}-\mathrm{PEP}1\right)}=1-{\left({\hat{\mathrm{RR}}}_{\;\left(\mathrm{CTRL}-\mathrm{PEP}1\right)}\right)}^{-1}$$

Therefore, 1 − (1/3.25) = 0.69. Thus, it is specifically estimated that treatment with PEP1 reduces the risk of death of chicks by 69% (*p* < 0.01), compared to the CTRL group. When using the one-sided interval estimation, with 95% confidence, for $\hat{\mathrm{PF}}$_CTRL-PEP1_:(6)$${\mathrm{CI}}_{95\%}\;\mathrm{to}\;{\hat{\mathrm{PF}}}_{\left(\mathrm{CTRL}-\mathrm{PEP}1\right)}=\left[0.26,1.00\right)$$

This implies that the reduction in mortality from the use of PEP1 is at least 26%, compared to the control group. The estimate of the protection factor is exerted by PEP2 treatment on the control group, and is given by:(7)$${\hat{\mathrm{PF}}}_{\left(\mathrm{CTRL}-\mathrm{PEP}2\right)}=1-{\left({\hat{\mathrm{RR}}}_{\;\left(\mathrm{CTRL}-\mathrm{PEP}2\right)}\right)}^{-1}$$

Therefore, 1 − (1/2.17) = 0.54. This result indicates that PEP2 treatment reduces the risk of mortality by 54% compared to the CTRL group (*p* = 0.04). Using the unilateral interval estimate for $\hat{\mathrm{PF}}$_CTRL-PEP2_, with 95% confidence:(8)$${\mathrm{CI}}_{95\%}\;\mathrm{to}\;{\hat{\mathrm{PF}}}_{\left(\mathrm{CTRL}-\mathrm{PEP}2\right)}=\left[0.04,1.00\right)$$

Consequently, the reduction in mortality from the use of PEP2 is at least 4% (more precisely, 4.05%), compared to the CTRL group.

The mortality rate after 5 dpi (days post-infection, 7th day of life) was zero for the chicks, subjected to all the treatments studied. Thus, the critical analysis period was concentrated from 2 dpi to 4 dpi per day of the experiment. With clear evidence, chicks treated with the ERCtx had a mortality rate of zero at 3 dpi (Figure 4). In relation to PEP2, there was only one death at 4 dpi, a fact that differs from the trend shown by treatments, and that may be caused by some eventuality, but it is not possible to conclude with certainty. There is a significant difference between the mortality rate of the CTRL group compared to the PEP1 and PEP2 treatments, on the second and third days of infection (*p* < 0.05). This shows that the antimicrobial peptide has the effect of reducing the risk of mortality already at the beginning of infection, when chickens ingest the microencapsulated peptide, which corresponds to the most acute phase of their mortality.

#### 3.2.2. Using a Power Test to Analyze Results

The one-sided 95% CI showed that the reduction, in percentage points, in the mortality rate from the control group to the PEP1 treatment group would have the lower limit (LL = 0.07 = 7%). This means that the existing population reduction would be at least 7%, which is highly satisfactory. For the reduction from CTRL to PEP2, it would be with LL = 0.02 = 2%, which is less than satisfactory. For this reason, it is not necessary to increase the sample size to demonstrate the efficacy of the Ctx(Ile^21^)-Ha antimicrobial peptide, when considering the performance of PEP1 treatment in reducing the mortality rate.

When assessing the sufficiency of the sample size, analysis with the Power Test can be included, which corresponds to the sensitivity of the test to reject the null hypothesis unequivocally, which can further improve test performance [24]. For this reason, a simulation was developed for different mortality rates lower than those obtained in the control treatment, according to the alternative hypothesis. The simulation for the sample size calculations was based on the estimates obtained, with a significance level of 5% and a power test of 80%, considered by Cohen [25] as acceptable. In Figure 5, the minimum number of sample units per treatment (for example, number of chicks used) is shown for the test to detect the reduction in mortality for proportions of 25 to 5%.

According to Figure 5, it is evident that the more the mortality proportion is reduced, the smaller the sample size necessary to detect the decrease in mortality, with a power of 80% and a significance of 5%. Mortality rates between 5 and 10% are known to require between 28 and 49 chicks per treatment, respectively. This result affirms that the 45 chicks used per treatment in this experiment would be sufficient to detect a reduction in mortality in the control treatment of the order of 20 percentage points or more, which occurs in the treatment with the lowest dose of peptide (PEP1). In another situation, if it is considered only for comparison (CTRL for PEP2), then a significance level greater than 5% would be adopted, or a power less than 80%, which could compromise the sensitivity of the test, or in general, test another four animals for each treatment to complete 49 animals per treatment, which would be unnecessary at this time, given the results found.

#### 3.2.3. Cecal Content

The data for the *S.* Enteritidis count in CFU mL^−1^ required transformation to a natural logarithm (Ln), to adapt the recommended model in ANOVA. The F test for treatment purposes showed statistical evidence (*p* < 0.02) that there is a difference between treatments. Likewise, the time effect of infection on the Ln count was strongly significant (*p* = 0.00), especially on the second day of infection (5 days of life). In addition to the main effects, the interaction effect was also investigated (*p* = 0.24).

This result shows that there is no evidence of an interaction between treatment and age; that is, the treatment showing the best performance is still PEP1, especially on the second day of infection, and after that this treatment alone, does not differ significantly from the others, as shown in Figure 6.

After ANOVA, a comparison test of the control treatment (CTRL) was performed against the other treatments (PEP1 or PEP2) using Dunnett’s test, which, in turn, showed statistical significance only between CTRL and PEP1 (*p* < 0.05).

#### 3.2.4. Chick Cloacal Swab

Results of the follow-up of the infection with the swab method of the *S.* Enteritidis inoculum (Figure 7) at a concentration of 10^9^ CFU mL^−1^, were subjected to the Chi-square test (*p* > 0.05), where the *p*-value found was 0.88. This result indicates that there is no evidence that the proportion of presence or absence is different between treatments. Therefore, based on this sample, it cannot be said that the treatment influences the absence or presence of the inoculum.

#### 3.2.5. Weighing of the Chicks

From the inferential perspective, the F-test for treatment purposes showed moderate evidence (*p* = 0.07) that there is a difference between treatments. The effect of age on weight, on the other hand, was strongly significant (*p* = 0.00) due to the intrinsic development of the body mass of the chicks, especially in the first weeks of life. In addition to these main effects, the presence of an interaction effect was also investigated (*p* = 0.74). This result shows that there is no evidence of interaction between treatment and age; that is, the effect of the treatments does not depend on age. These results could be illustrated using a boxplot (Figure 8).

## 4. Discussion

Due to the absence of similar studies for parameter estimations necessary for the calculations, the minimum number of chicks required to demonstrate the efficacy of the treatment in reducing mortality was not determined. This is in agreement with Montgomery and Runger [20], when they report on the minimum conditions to use the approximation of the binomial by normal distribution, which is necessary when calculating the sample size that involves proportional estimation. The approximation conditions are: *np* > 5 and *np*(1 − *p*) > 5, where “*n*” is the sample size, “*p*” is the proportion of the event of interest: in this case, the death of the chicks. These same authors note that, in general, a better approximation is given for large samples (*n* > 40).

It is known that the mortality rate caused by *S.* Enteritidis is low [26]. However, infection with this bacterium weakens the immune system and causes collateral damage that affects nutrient absorption [27]. Consequently, other bacteria can act as opportunists, colonizing the intestine and causing poultry death. Systemic poultry infection would be linked to the influence of the flagella in some *Salmonella* sp. serovars [26]. Thus, when it comes to newborn chickens (up to 5 dpi), some studies suggest the use of immune stimulators to reduce the mortality rate [27]. Another study affirms that the BT peptide was able to promote mRNA transcription for Toll-like receptors (TLR), responsible for producing a pro-inflammatory response with cytokines and, consequently, activating the immune response. They also highlighted that the use of AMPs in the first four days of life is important, and that its best application would be orally [14].

A recent study indicates that AMPs have the same bacteriostatic potential compared to conventional antibiotics, and they also increased the content of white blood cells, making them an excellent replacement alternative for bio-sustainable poultry production, even better than other natural components [28]. In addition, due to its rapid way of acting against bacteria, the risk of acquiring or generating bacterial resistance is minimal, since the main target of lytic AMPs are the plasmatic membranes [19].

A study with essential oils in a combination of *Syzygium aromaticum* and *Cinnamomum zeylanicum* showed antimicrobial activities against *S.* Enteritidis and *S. typhimurium* (0.322–0.644 mg mL^−1^ and 0.644–1.289 mg mL^−1^, respectively). However, these MIC values were very low compared (9.32 and 37.30 µg L^−1^, respectively) to those reached by the Ctx(Ile^21^)-Ha antimicrobial peptide. It is worth mentioning that in vitro studies revealed that the microencapsulated Ctx(lle^21^)-Ha peptide presented antimicrobial activity with pathogens from the poultry sector such as *Salmonella* Enteritidis, *Salmonella* Typhimurium and *Escherichia coli* [19].

MccJ25 is a highly studied cyclic recombinant AMP, presenting a broad bactericidal spectrum, mainly against *Salmonella* sp. [29] This AMP showed that its function is not only to eliminate the bacteria and improve the fecal microbiota, but also to influence intestinal morphology by improving texture and reducing inflammation after infection [30]. Moreover, bacteriocins are an AMP group studied for use in the poultry and swine industry. They are produced by some bacteria (the majority by Gram-positive) and present interesting effects by reducing the content of pathogenic bacteria, such as *Salmonella* sp. and *Campylobacter jejuni* [31]. Other bacteriocins were studied to reduce *Salmonella* in broilers using a dose of 2.5 g kg^−1^, exhibiting an increase in the weight of the chickens and a slight bacterial decrease [32]. Unlike our results, which could be due to the minimum dose used in this experiment, which is thousands of times less than the reference, our work used mg (miligrams) of peptide instead of g (grams), used by the reference mentioned. However, other research indicates that the application of swine intestine [33] as a food supplement influences positively broilers’ weight, as well as an increased villus height [34]. As an explanation, the hypothesis is if Ctx(Ile^21^)-Ha peptide concentration is incremented, it will be possible to visualize a more pronounced increase in weight in chicks. However, for a pilot in vivo experiment, the Ctx(Ile^21^)-Ha peptide microcapsules showed very promising and interesting results.

Studies based on previous AMP-encapsulates, such as bacteriocins, were protected using polyvinylpyrrolidone as an encapsulating material and showed promising results due to decrease content of *C. jejuni*. However, the AMP dose employed was very high compared with our encapsulated products (500 mg of AMP per kg^−1^ of poultry feed) [35]. Our product is encapsulated based on sodium alginate, which is a cheap and more biocompatible polymer [36]. Sodium alginate encapsulations loaded with specific phages (f3αSE) for *S.* Enteritidis have been made, achieving up to 80% protection of viability of this product at a gastric pH [37]. This confirms and corroborates that our developed microcapsules can satisfactorily contain and protect the Ctx(Ile^21^)-Ha AMP, as they also have an additional protective layer with HPMCP. This polymer is degraded only at intestinal pH used as a drug targeting, and which also protects microparticles and AMP from mechanical processes that could be subjected to in manufacturing or by the gizzard [11,19].

Consequently, this study showed a decrease in mortality rate in first days of life, which certifies the success of the encapsulated and coated antimicrobial peptide. Values of anti-*S.* Enteritidis activity in vivo did not make a significant difference. However, there is an interesting result in first days of life, decreasing the total count. This could be due to the relationship between the amount of microcapsule ingested and the total volume of the intestine, and that each time the chicks grew, this relationship would be more different. Therefore, it is suggested that further experiments employing higher doses can be performed to achieve a total bacterial decrease.

## 5. Conclusions

The in vivo analyses allow us to conclude that the antimicrobial peptide Ctx(lle^21^)-Ha presented positive, significant and promising results in relation to the reduction in younger chicken’s mortality and the bacterial count, mainly for PEP1 treatment, where there is a 69% reduction in the risk of death. Regarding the weight of the chickens, in two doses of antimicrobial peptide used, there was a significant difference between treatments and this result shows that there is no evidence of interaction between treatment and age; that is, the effect of the treatments does not depend on age. Finally, it is concluded that there is a potential effect of the microencapsulated-coated antimicrobial peptide Ctx(lle^21^)-Ha in poultry, which enables the application of the peptide by using a very low mass compared to other studies in the literature.

## 6. Patents

The present methodology and application developed was deposited in the National Institute of Intellectual Property (INPI BR1020200220489), which is protected throughout the Brazilian territory.

## Figures and Tables

**Figure 1 antibiotics-10-00616-f001:**
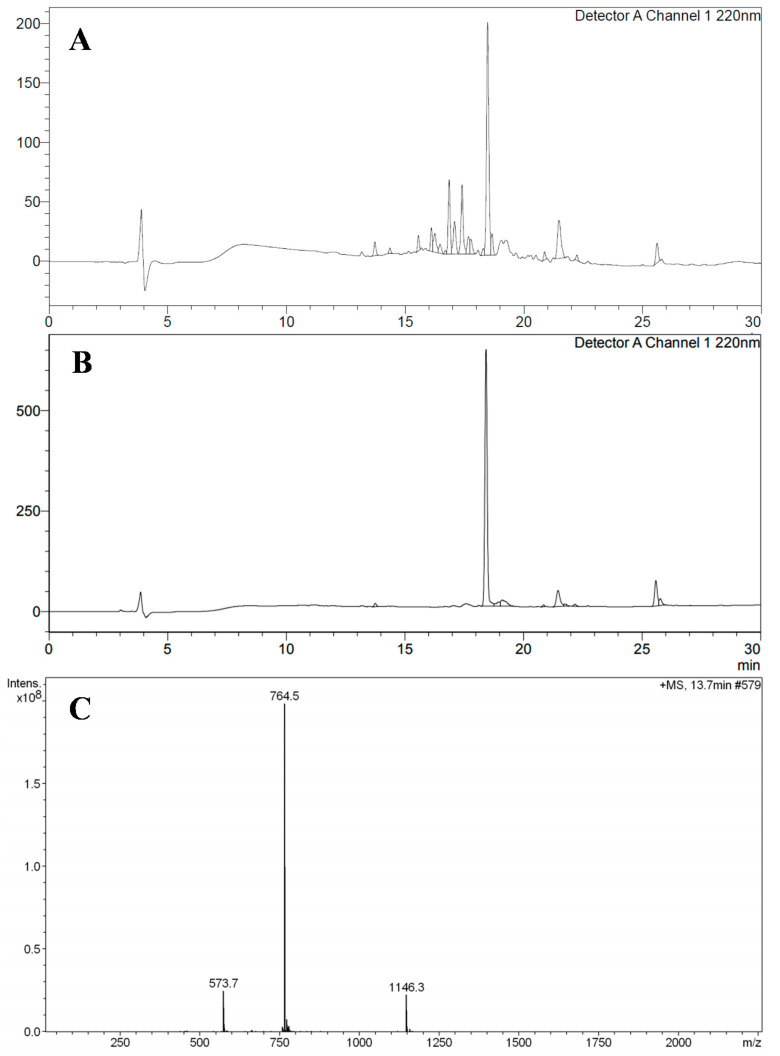
(**A**): Chromatographic profile of crude Ctx(Ile^21^)-Ha peptide by HPLC at 220 nm. (**B**): Chromatographic profile of purified Ctx(Ile^21^)-Ha peptide by HPLC at 220 nm. (**C**): Mass spectra of Ctx(Ile^21^)-Ha peptide, confirming the correct obtaining.

**Figure 2 antibiotics-10-00616-f002:**
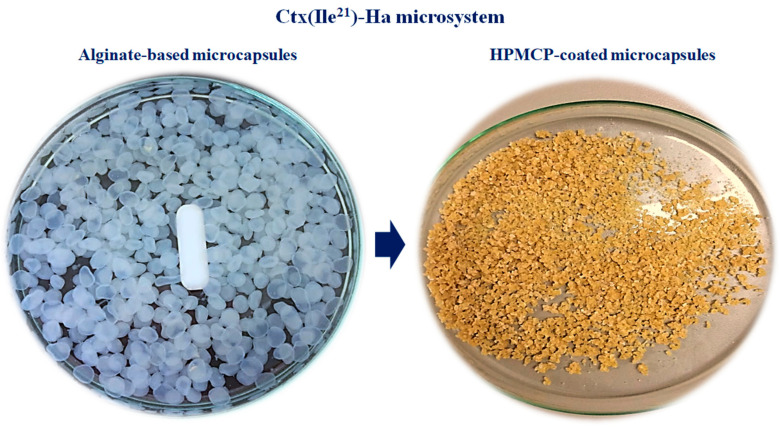
Microcapsules obtained after ionic gelation and fluidized bed.

**Figure 3 antibiotics-10-00616-f003:**
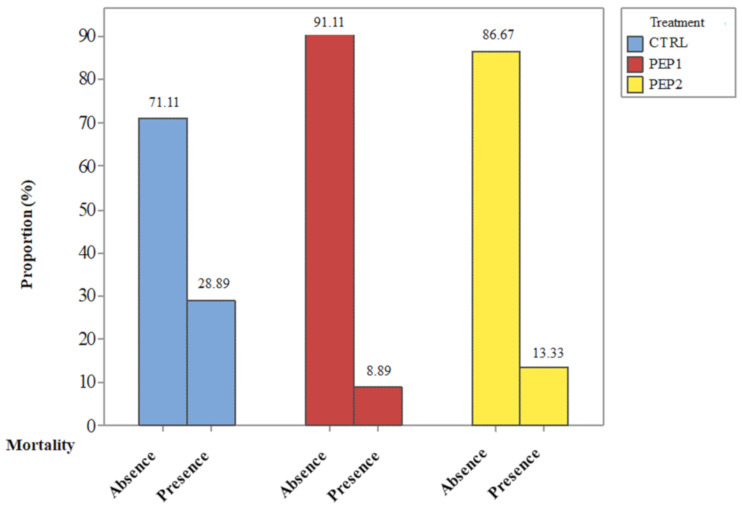
Percentage distribution of mortality resulted from the in vivo treatments analyzed.

**Figure 4 antibiotics-10-00616-f004:**
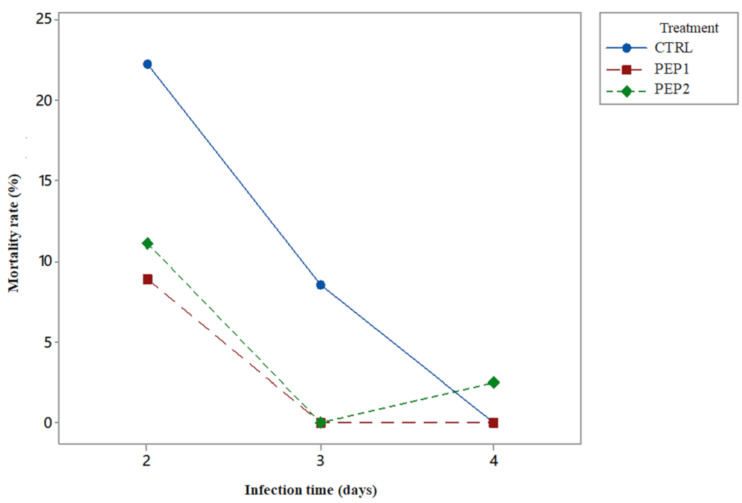
Mortality rate as a function of time of infection.

**Figure 5 antibiotics-10-00616-f005:**
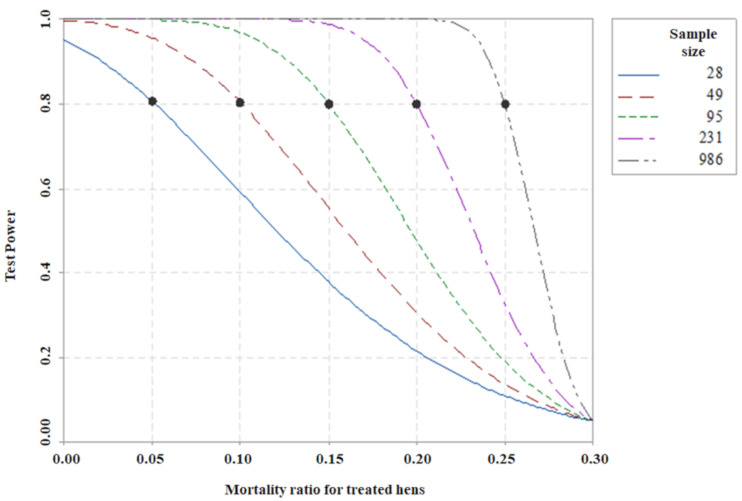
Calculation of the theoretical sample size to demonstrate the test power obtained in this experiment.

**Figure 6 antibiotics-10-00616-f006:**
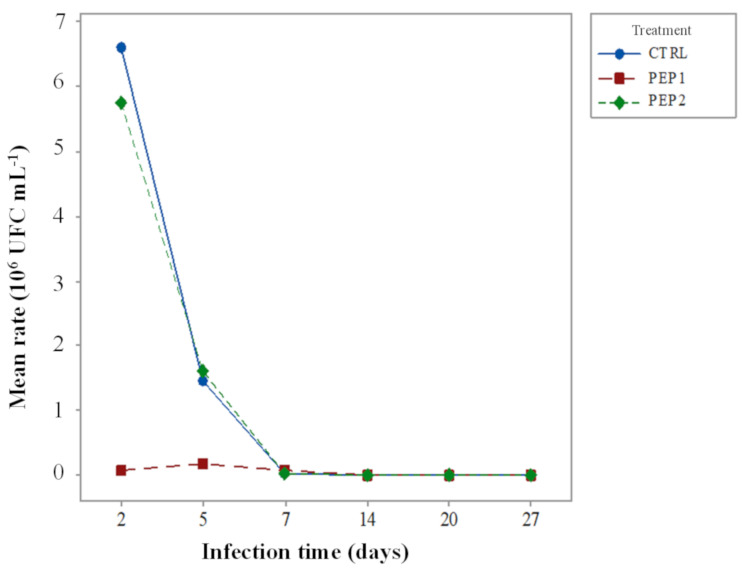
Mean count rate of *S.* Enteritidis, which is dependent on the treatments and time of infection.

**Figure 7 antibiotics-10-00616-f007:**
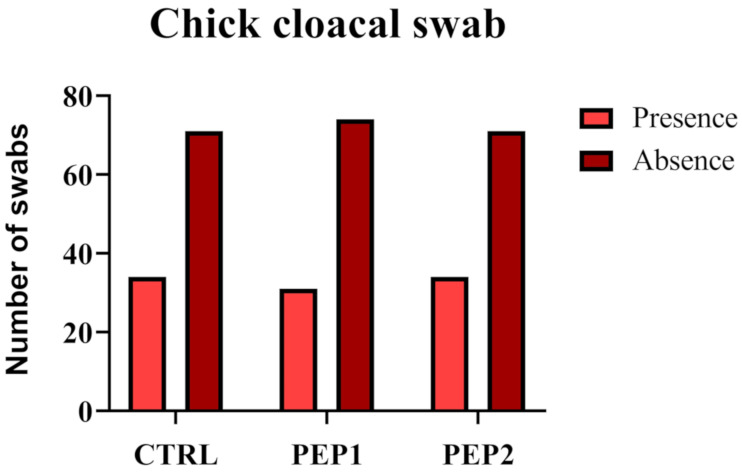
Evaluation of the effect of antimicrobial peptide on *Salmonella* Enteritidis fecal excretion during 28 days.

**Figure 8 antibiotics-10-00616-f008:**
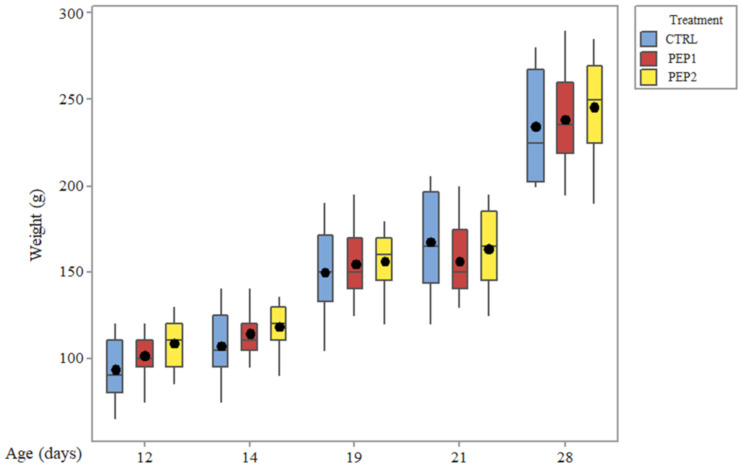
Boxplot of weight distribution according to the treatments and infection time to which the chicks were subjected.

## Data Availability

Not applicable.

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
