# Peer review of "HPMCP-Coated Microcapsules Containing the Ctx(Ile21)-Ha Antimicrobial Peptide Reduce the Mortality Rate Caused by Resistant Salmonella Enteritidis in Laying Hens"

_antibiotics, 2021, doi:10.3390/antibiotics10060616_

Round 1

Reviewer 1 Report

The authors concluded that there is a potential effect of the Ctx(lle21)-Ha in poultry, which enable the application of the peptide using very low mass, compared to other studies in literature. However, the number of chicken used to study the effect of Ctx(lle21)-Ha on body weight is limited (N=15 for each group). From the results we also can see that Ctx(lle21)-Ha have no significant effect on body weight of chicken. It is difficult to infer Ctx(lle21)-Ha will have a promising use in poultry, compared to other antimicrobial peptide. Besides, the authors showed that the Ctx(lle21)-Ha had no effect on the bacterial count from 7 dpi to 27, I think the dead of chicken at the first days of infection would influence the results. In my opinion, I do not think this manuscript can be accepted in the Journal with high impact fact.

Material and Methods

  1. 1. Chemical Reagents. “and the other chemical reagents were obtained in HPLC grade (Sigma-Aldrich Co., St Louis, MO, USA). ” The description in this section was repeated.
  2. 4 In vivo experiment in chicks. “Although, due to the absence of similar studies for parameters estimation….. approximation is given for large samples (n > 40).” This should belong to discussion part, and it should not be included in the materials and methods part.
  3. The sex of the sampled chicks in this experiment should be added.
  4. “was grown in NB for 24 h at 37 °C.” The full name of NB should be added.

Results

  1. 2.1. Post-inoculation mortality. Table 1. The word “Mortality” in Table 1 is not proper. Because the content also included the chicken alive.
  2. The authors described that “?̂ ????= 32/45 = 28.89%”. It should be ?̂ ????= 13/45 = 28.89%

Author Response

Tupã, 13th May, 2021.

To

Ms. Tabita Hent   
Assistant Editor

MDPI Cluj
Str Avram Iancu 454, Floresti, Cluj Romania
Antibiotics Editorial Office

Please find enclosed the revised version of the research manuscript “HPMCP-Coated Microcapsules Containing the Ctx(Ile21)-Ha Antimicrobial Peptide Reduce the Mortality Rate Caused by Resistant Salmonella Enteritidis in Poultry", submitted as a research article to Antibiotics. We deeply appreciated the careful Reviewers’ revision in order to contribute for manuscript improvements and the paper publication. As suggested by Reviewers, all the points were carefully revised one by one and corrected in the manuscript (terms and phrases added are highlighted in blue and words excluded are shown in red). Our responses for the reviewers' comments are shown below.

REVIEWER #1:

            The authors concluded that there is a potential effect of the Ctx(lle21)-Ha in poultry, which enable the application of the peptide using very low mass, compared to other studies in literature. However, the number of chicken used to study the effect of Ctx(lle21)-Ha on body weight is limited (N=15 for each group). From the results we also can see that Ctx(lle21)-Ha have no significant effect on body weight of chicken. It is difficult to infer Ctx(lle21)-Ha will have a promising use in poultry, compared to other antimicrobial peptide. Besides, the authors showed that the Ctx(lle21)-Ha had no effect on the bacterial count from 7 dpi to 27, I think the dead of chicken at the first days of infection would influence the results. In my opinion, I do not think this manuscript can be accepted in the Journal with high impact fact.

Author reply: We highly appreciated the Reviewer’ comments. The main point of the present study is the great decrease of mortality rate in laying hens. Specially, the PEP1 treatment showed a very high reduction (69%) in the risk of death (statistically significant), which evidences the clear effectiveness of the Ctx(Ile21)-Ha antimicrobial peptide combined with microencapsulation technique for coating/protection in the gastrointestinal tract. Unfortunately, the objective of this study was not comparing the biological therapeutic action among different AMPs. In this way, to the best of our knowledge, this work presents an innovative approach, which shed light on how it can become feasible the oral application of a natural molecule, in very small quantities, to be applied in laying hens, aiming the effectiveness against important pathogens that can affect the poultry production section.

            Regarding the comment “the authors showed that the Ctx(lle21)-Ha had no effect on the bacterial count from 7 dpi to 27, I think the dead of chicken at the first days of infection would influence the results”, according to the second paragraph of the Discussion section in the manuscript: “It is known that the mortality rate caused by S. Enteritidis is low [23]. However, infection with this bacterium weakens the immune system and causes collateral damage that affects nutrient absorption [24]. Consequently, other bacteria can act as opportunists, colonizing the intestine and causing poultry death.”

            Following the technical instructions of Hy-Line brown laying hens, the maximum expected mortality rate on the first week of life is 0,75%, which decreases exponentially over the weeks due to the immune system development. The laying hens used in this experiment were challenged with Salmonella in the first days of life (when the immune system is still very weak), which caused a high mortality rate, mainly in the animals of control group (untreated). With the normal development and effective performance of the laying hens’ immune system after the first week of life, it is believed that there was also a decrease in bacterial count after this period (7 to 27 dpi), since the mortality rate caused normally by S. Enteritidis is low [reference 23], under conditions in which these animals are not challenged by this pathogen or when their immune system is already developed, or developing after birth. Moreover, the present study showed that the greater protection promoted by the Ctx(Ile21)-Ha microencapsulation was obtained with less costs, which is very important approach for an advance industrial application step. As the production of high quantities of peptide (large scale) is not possible at the moment, the initial tests were done on animals of younger age categories, due to less food intake (which was directly related to drug requirements and certain less peptide mass for mixture in feed) to verify its efficacy, which was demonstrated by the statistical analysis of the present paper.

 Material and Methods

  1. Chemical Reagents. “and the other chemical reagents were obtained in HPLC grade (Sigma-Aldrich Co., St Louis, MO, USA).” The description in this section was repeated.

Author reply: Thank you for your comment. The repeated phrase was removed from the section.

      2. 4 In vivo experiment in chicks. “Although, due to the absence of similar studies for parameters estimation… approximation is given for large samples (n > 40).” This should belong to discussion part, and it should not be included in the materials and methods part.

Author reply: Thank you for your suggestion. This part was removed from the Materials and Methods and included in the Discussion section.

      3. The sex of the sampled chicks in this experiment should be added.

Author reply: Thank you for your observation. This information was included in the “2.4. In vivo experiment in chicks subsection from Material and Methods.

      4. “was grown in NB for 24 h at 37 °C.” The full name of NB should be added.

Author reply: Thank you for your suggestion. The full name of the medium was added to the manuscript. We highlight here that all the broth full names are described in the “Chemical Reagents” subsection.

Results

  1. 2.1. Post-inoculation mortality. Table 1. The word “Mortality” in Table 1 is not proper. Because the content also included the chicken alive.

Author reply: Thank you for your comment. The Table 1 was excluded by suggestion of Reviewer #3, since Figure 3 properly explains the post-inoculation mortality risk.

      2. The authors described that “?̂ ????= 32/45 = 28.89%”. It should be ?̂ ????= 13/45 = 28.89%

Author reply: Thank you for your suggestion. The number was corrected in the manuscript.

Therefore, we hope the modifications that were carefully performed and rewritten in the manuscript can make it suitable for publishing in Antibiotics. Therefore, we expected that our corrections satisfy the Reviewer and Editors’ points.

            Sincerely,

Prof. Eduardo Festozo Vicente, Ph. D.

      Correspondent Author

Reviewer 2 Report

The work is very interesting and brings new elements to the current state of knowledge regarding the use of antimicrobial peptide in poultry.

The purpose of the work is clearly stated. The conclusions of the conducted research are clear and result from the obtained research results.

The material used for the research is sufficient, the research methods have been selected appropriately. The layout of the tables and figures is correct. The differences between the groups were marked correctly.

Discussing the results against the background of other authors is very detailed.

The publications cited by the authors of the article are well selected. For the most part, the authors refer to the latest knowledge published in renowned scientific journals.

Author Response

Tupã, 13th May, 2021.

To

Ms. Tabita Hent   
Assistant Editor

MDPI Cluj
Str Avram Iancu 454, Floresti, Cluj Romania
Antibiotics Editorial Office

Please find enclosed the revised version of the research manuscript “HPMCP-Coated Microcapsules Containing the Ctx(Ile21)-Ha Antimicrobial Peptide Reduce the Mortality Rate Caused by Resistant Salmonella Enteritidis in Poultry", submitted as a research article to Antibiotics. We deeply appreciated the careful Reviewers’ revision in order to contribute for manuscript improvements and the paper publication. As suggested by Reviewers, all the points were carefully revised one by one and corrected in the manuscript (terms and phrases added are highlighted in blue and words excluded are shown in red). Our responses for the reviewers' comments are shown below.

REVIEWER #2:

            The work is very interesting and brings new elements to the current state of knowledge regarding the use of antimicrobial peptide in poultry.

            The purpose of the work is clearly stated. The conclusions of the conducted research are clear and result from the obtained research results.

            The material used for the research is sufficient, the research methods have been selected appropriately. The layout of the tables and figures is correct. The differences between the groups were marked correctly.

            Discussing the results against the background of other authors is very detailed.

            The publications cited by the authors of the article are well selected. For the most part, the authors refer to the latest knowledge published in renowned scientific journals.

Author reply: We deeply appreciate the positive comments by the Reviewer.

            Sincerely,

Prof. Eduardo Festozo Vicente, Ph. D.

Correspondent Author

Reviewer 3 Report

The objective of the authors showed that the microen-capsulated Ctx(Ile 21 )-Ha antimicrobial peptide can be an interesting and promising option in the substitution of conventional antibiotics . The manuscript is well written, and the authors followed what have been used in the literature for this type of study. However, I highly recommend revised the manuscript. Therefore, I recommend the publication of this paper after proposed corrections:

  1. The mechanism of antimicrobial peptides is similar to that of antibiotics, whether long-term use will cause bacterial resistance?
  2. please clarify the chicks used in this study, laying hens? If laying hens were used, please change “poultry” to “laying hens” in the title
  3. In the result part, the data needn’t be presented in both figures and tables, for example table 1 and figure 3, table 2 and figure 8
  4. Please clarify how many replicates of one treatment?
  5. Please clarify what type of feed was used in this experiment, granule or powder?
  6. Whether the granulation process of feed affects the effect of the peptide?
  7. In this study, in vitro experiment was not involved, please delete this sentence in the conclusion part “In vitro studies revealed that the microencapsulated Ctx(lle 21 )-Ha peptide presented antimicrobial activity with pathogens from poultry sector such as Salmonella Enteritidis, Salmonella Typhimurium and Escherichia coli.” Or you can also put in the discussion part but it must be used as a reference

    The objective of the authors showed that the microen-capsulated Ctx(Ile 21 )-Ha antimicrobial peptide can be an interesting and promising option in the substitution of conventional antibiotics . The manuscript is well written, and the authors followed what have been used in the literature for this type of study. However, I highly recommend revised the manuscript. Therefore, I recommend the publication of this paper after proposed corrections:

    1. The mechanism of antimicrobial peptides is similar to that of antibiotics, whether long-term use will cause bacterial resistance?
    2. please clarify the chicks used in this study, laying hens? If laying hens were used, please change “poultry” to “laying hens” in the title
    3. In the result part, the data needn’t be presented in both figures and tables, for example table 1 and figure 3, table 2 and figure 8
    4. Please clarify how many replicates of one treatment?
    5. Please clarify what type of feed was used in this experiment, granule or powder?
    6. Whether the granulation process of feed affects the effect of the peptide?
    7. In this study, in vitro experiment was not involved, please delete this sentence in the conclusion part “In vitro studies revealed that the microencapsulated Ctx(lle 21 )-Ha peptide presented antimicrobial activity with pathogens from poultry sector such as Salmonella Enteritidis, Salmonella Typhimurium and Escherichia coli.” Or you can also put in the discussion part but it must be used as a reference

Author Response

Tupã, 13th May, 2021.

To

Ms. Tabita Hent   
Assistant Editor

MDPI Cluj
Str Avram Iancu 454, Floresti, Cluj Romania
Antibiotics Editorial Office

Please find enclosed the revised version of the research manuscript “HPMCP-Coated Microcapsules Containing the Ctx(Ile21)-Ha Antimicrobial Peptide Reduce the Mortality Rate Caused by Resistant Salmonella Enteritidis in Poultry", submitted as a research article to Antibiotics. We deeply appreciated the careful Reviewers’ revision in order to contribute for manuscript improvements and the paper publication. As suggested by Reviewers, all the points were carefully revised one by one and corrected in the manuscript (terms and phrases added are highlighted in blue and words excluded are shown in red). Our responses for the reviewers' comments are shown below.

REVIEWER #3:

            The objective of the authors showed that the microencapsulated Ctx(Ile21)-Ha antimicrobial peptide can be an interesting and promising option in the substitution of conventional antibiotics. The manuscript is well written, and the authors followed what have been used in the literature for this type of study. However, I highly recommend revised the manuscript.

Author reply: We deeply thank your comments. All the suggestions were accepted and corrected in the manuscript. Point-by-point comments can be seen below.

Therefore, I recommend the publication of this paper after proposed corrections:

  1. The mechanism of antimicrobial peptides is similar to that of antibiotics, whether long-term use will cause bacterial resistance?

Author reply: Thank you for the comment. In page 14, a brief paragraph explaining this question was added.

       2. Please clarify the chicks used in this study, laying hens? If laying hens were used, please change “poultry” to “laying hens” in the title

Author reply: Thank you for the comment. In fact, we had used laying hens as study object. Therefore, the title was corrected, replacing the word “poultry” to laying hens”.

        3. In the result part, the data needn’t be presented in both figures and tables, for example table 1 and figure 3, table 2 and figure 8.

Author reply: Thanks for the comment. Following the Reviewer’ suggestion, all tables mentioned were excluded from the manuscript.

        4. Please clarify how many replicates of one treatment?

Author reply: Thanks for your question. The number of replicates used was described and clarified in the “2.4 In vivo experiment in chicks” subsection.

        5. Please clarify what type of feed was used in this experiment, granule or powder?

Author reply: Thanks for your question. The type of feed used in the experiments was described and clarified in the “2.4 In vivo experiment in chicks” subsection, on page 4.

          6. Whether the granulation process of feed affects the effect of the peptide?

Author reply: Thanks for your question. The alginate-based microparticles loaded with the Ctx(Ile21)-Ha antimicrobial peptide have an HPMCP coating, which protects from some mechanical processes both in granulation and in the digestion produced by the gizzard. This explanation was also added in the Discussion section.

        7. In this study, in vitro experiment was not involved, please delete this sentence in the conclusion part “In vitro studies revealed that the  microencapsulated Ctx(lle21)-Ha peptide presented antimicrobial activity with pathogens from poultry sector such as Salmonella Enteritidis, Salmonella Typhimurium and Escherichia coli.” Or you can also put in the discussion part but it must be used as a reference

Author reply: Thanks for your observation. This statement was removed from the Conclusion section, as suggested.

We hope the modifications that were carefully performed and rewritten in the manuscript can make it suitable for publishing in Antibiotics. Therefore, we expected that our corrections satisfy the Reviewer and Editors’ points.

           Sincerely,

Prof. Eduardo Festozo Vicente, Ph. D.

Correspondent Author

Round 2

Reviewer 1 Report

My concerns have been responded and the authors changed the conclusion about the effect of  Ctx(Ile 21 )-Ha on body weight. I agree that this manuscript will be published in the journal.

Reviewer 3 Report

I content the response of the author and recommend to publish this study.